# Economic assessment of potential changes to essential medicines for diabetes in Uganda

Atousa Bonyani[1]*, Tracy K. Lin[2], Ambrose Jakira[3], Isaac D. Kimera[4], Martin Muddu[5], Jeremy I. Schwartz[6], James G. Kahn[7]

1 Institute for Global Health Sciences, University of California, San Francisco, San Francisco, California, United States of America, 2 Department of Social and Behavioral Sciences, Institute for Health & Aging, University of California, San Francisco, San Francisco, California, United States of America, 3 Department of Pharmaceuticals and Natural Medicines, Ministry of Health, Kampala, Uganda, 4 Makerere University Joint AIDS Program, Kampala, Uganda, 5 Makerere University School of Medicine, Kampala, Uganda, 6 Section of General Internal Medicine, Yale School of Medicine, New Haven, Connecticut, United States of America, 7 Philip R. Lee Institute for Health Policy Studies, University of California, San Francisco, San Francisco, California, United States of America

* atousa.bonyani@ucsf.edu

## Abstract

### Background

The World Health Organization suggests selecting essential medicines based on availability, accessibility, affordability, cost-effectiveness, and safety and efficacy. We examined Oral Hypoglycemic Agents (OHAs) for type 2 diabetes mellitus in the Essential Medicines and Health Supplies List for Uganda (EMHSLU), assessed all criteria, and proposed alternative medicines to improve cost-savings and health outcomes.

### Methods

We conducted a literature review followed by budget impact analysis (BIA) to evaluate potential cost-savings from refining EMHSLU. In our literature review, we identified metrics for measuring availability, accessibility, and affordability of medicines and evaluated evidence on cost-effectiveness, safety, and efficacy of OHAs. According to the country context and available data, we used applicable methods to analyze five classes of OHAs available in Uganda. Since the government covers essential medicines in the public sector based on BIA, we assessed affordability for the government by examining potential savings in two scenarios (#1: shifts within the EMHSLU; #2: use of non-EMHSLU drugs).

### Results

OHAs added to the EMHSLU 2023 are less available and accessible compared to OHAs in EMHSLU 2016. BIA scenario 1 revealed that the complete replacement of

**Data availability statement:** All relevant data are within the manuscript and its Supporting Information files.

**Funding:** The author(s) received no specific funding for this work.

**Competing interests:** The authors have declared that no competing interests exist.

glimepiride 2 mg and gliclazide 80 mg with glibenclamide 5 mg could save the government up to USD 2.65 million per year. Additionally, scenario 2 showed that full replacement of dapagliflozin 10 mg and glimepiride 4 mg instead of their lower doses can save up to United States Dollar (USD) 230,000 and USD 600,000 per year, respectively. A review and adaptation of a published cost-effectiveness analysis highlights that sitagliptin is a cost-effective OHA in Uganda with an estimated Incremental Cost Effectiveness Ratio of $6,132/Quality-Adjusted LifeYear.

## Conclusion

The findings suggest several potential type 2 diabetes oral medication substitutions which would reduce costs.

## Introduction

Essential medicines are those that address the most critical healthcare needs of the population, considering factors such as disease prevalence, public health importance, evidence of effectiveness and safety, and cost-effectiveness [1]. As defined by the World Health Organization (WHO), access to essential medicines comprises five fundamental aspects: (1) availability at the national level, (2) geographic accessibility, (3) affordability, (4) cost-effectiveness, and (5) overall safety and efficacy [1,2]. WHO has put forth models of Essential Medicines List (EML) to identify medications that most effectively address key healthcare requirements [2]. Per WHO guidance, countries should tailor the EML to their specific needs by considering disease prevalence, the healthcare system's ability to efficiently distribute medications, purchase costs, special requirements by sub-populations, and the preferences of patients and healthcare providers [3,4]. Concomitantly, factors such as demographic features and Gross Domestic Product (GDP) per capita may influence this adaptation process [5].

Countries have adopted varied timelines and criteria for the selection of their National Essential Medicines List (NEML) [6–8]. While some update their lists every two years, others may take more than five years between revisions, and in some cases, there is no documented revision schedule [6]. High-income countries mainly described their approach to selecting NEML as a process grounded in evidence, prioritizing quality, safety, and efficacy of a medication [7]. In contrast, many lower-income countries rely primarily on experience and subjective judgment, with scientific evidence having a lesser role in decision-making [8]. The lack of a unified approach to refining the NEML highlights the need for a systematic method for this critical task. This process should be based on routine, evidence-based, and transparent assessments of NEMLs and should ensure that NEMLs effectively address the country's need to reach Universal Health Coverage (UHC) [7,9]. Changes to NEMLs can have several implications, including impacts on medicine availability, affordability, and prescribing practices that should be anticipated into the NEMLs revision process [10].

There are several existing instruments, tools, and approaches for the adaptation of NEML. However, there is no tool that comprehensively assesses all the selection

criteria suggested by WHO [11]. Some countries adopt Health Technology Assessment (HTA) as a tool to review cost-effectiveness, clinical evidence, and ethical aspects of the new technologies as main policy-making elements [12]. However, HTA typically does not consider the availability, affordability, and accessibility of medicines, and there is no evidence that utilizing HTA will enhance availability, affordability, and cost savings [11]. Therefore, an inclusive system encompassing the availability, accessibility, and affordability of medicines in addition to the traditional HTA would be most helpful for policymakers in formulating their NEML.

To evaluate and identify an approach that can circumvent current challenges with EML, this study analyzed the Oral Hypoglycemic Agents (OHA) for the treatment of type 2 diabetes mellitus (T2DM) in NEML and explored how a structured approach might contribute to the refinement of EML. T2DM is selected as a high-burden disease worldwide due to its high prevalence, mortality, and morbidity, and we focused on OHAs as they represent a significant portion of the DM management and control medication budget [13,14]. We leveraged the Essential Medicines and Health Supplies List for Uganda (EMHSLU) as a well-structured NML, which provides a strong foundation assessment for our case study. With a prevalence of 4.6% in 2021 and an estimated 48% of patients remaining undiagnosed, diabetes is considered a high-burden disease in Uganda [15,16]. EMHSLU is developed by the government to ensure the population's health needs are met through public health facilities and indicates which medicines are designated to be provided at each level of care [17]. The current process for assigning a health product to EMHSLU involves evaluation of the clinical efficacy, cost-effectiveness, public health relevance, and alignment with national guidelines by a national task force before final approval by the Ministry of Health [18]. According to official reports from 2018, nearly 45% of total health facilities in Uganda were governmental, 15% were private-not-for-profit and 40% were private-for-profit [19]. Although health services are free for people in the public sector, the lack of accessibility to medicines and health services in this sector often leads people to use the private sector which results in higher out-of-pocket payments and affordability issues [20,21]. Our overarching objective is to evaluate mechanisms to optimize NEMLs. To contribute to this objective, we conducted a systematic review of published evidence to assess the availability, accessibility, affordability, cost-effectiveness, safety, and efficacy of OHAs for diabetes in Uganda, with the aim of optimizing EMHSLU and ultimately enhancing diabetes care and health outcomes in the country.

## Methods

WHO advises countries to consider availability, accessibility, affordability, cost-effectiveness as well as clinical profiles (safety and efficacy) of medicines when selecting their NEML [1]. Taking WHO recommendations into consideration, we employed 1) a literature review to identify metrics used in peer-reviewed studies for measuring each criterion, and then to select context-appropriate metrics for Uganda, followed by 2) a Budget Impact Analysis (BIA) to analyze available data and assess potential cost-savings from refining the EMHSLU.

### Data collection and analysis

**Availability, accessibility, affordability, cost-effectiveness, and clinical profiles (safety and efficacy) of medicines.** We used the following definitions to guide our selection criteria and search. **Availability** refers to the presence and supply of needed dosage forms of quality medicines at the national or warehouse level and in sufficient quantity. This element requires a strong regulatory system to issue production or importation licenses. **Accessibility**, also known as geographical accessibility and at the health facility levels, examines the extent to which medicines are available in different geographical regions, influenced by health facility distribution, supply chain system performance, and procurement systems [22]. **Affordability** is defined through different dimensions: the product value dimension considers cost-effectiveness and Quality Adjusted Life Year (QALY), the healthcare system perspective dimension focuses on optimizing the health system budget using budget impact analysis (BIA), and the family perspective dimension considers minimum worker wages or Gross Domestic Product (GDP) [23]. **Cost-effectiveness** for medicines evaluates the costs and health outcomes associated with pharmaceutical interventions, comparing the relative expenses and benefits of

different medications to determine which provides the best value for money [24]. **Safety** pertains to considering side effects and potential harm associated with the medicines, while **Efficacy** reviews the ability of medicines to produce the desired health outcomes [25].

To identify metrics for measuring availability, accessibility, and affordability, we conducted a literature review of peer-reviewed papers in PubMed, Scopus, and Google Scholar in March 2024 with the keywords: Essential Medicines, Availability, Accessibility, Affordability, Biguanides, Sulfonylureas, Thiazolidinediones, SGLT2 inhibitors, and DPP-4 inhibitors. We then listed the measurement methods published in the literature for the availability, accessibility, and affordability of medicines to examine which of them are applicable to our case study in Uganda. For cost-effectiveness, safety, and efficacy, we gathered published results of OHA options within each drug class to identify the best options for selection.

**Characterize EMHSLU.** To conduct a case study for Uganda, we obtained up-to-date data directly from the Ministry of Health (MoH) of Uganda [26]. This information allowed us to characterize the EMHSLU with respect to criteria outlined by identified methods and metrics. We focused on five classes of OHAs available in the country: biguanides, sulfonylureas, thiazolidinediones, SGLT-2 inhibitors, and DPP-4 inhibitors, and reviewed published evidence on cost-effectiveness, safety, and efficacy. Finally, based on our analysis, we 1) proposed programmatic and managerial actions to enhance the availability, accessibility, and affordability of medicines in the current EMHSLU, and 2) recommended changes to the EMHSLU drug selections based primarily on reduced costs, as well as cost-effectiveness and medicines' clinical profiles.

To further characterize EMHSLU, we conducted a literature review in PubMed, Scopus, and Google Scholar as well as a search of gray literature in May 2024 using the following keywords: availability, accessibility, affordability, Uganda, Supply Chain, Diabetes Medicines, Oral Hypoglycemic Agents. In addition to the data obtained from published peer-reviewed papers, we gathered information from the National Regulatory Authority (NRA) and MoH websites, EMHSLU 2016 and 2023, Uganda Clinical Guidelines 2023, routine health facility reporting database, and pharmaceutical importers' reports. Based on the resources available, we reviewed the following data for each criterion:

**Availability:** Using the NRA website, we extracted information regarding granted licenses for OHAs. Of those medicines that received NRA licenses, we analyzed local production and import status as well as the volume of imports of medicines in EMHSLU 2016 and 2023 in a two-year period (from July 1, 2021, to June 30, 2023) according to MoH and NRA reports. Lastly, by contacting the importers and distributors, we excluded from our analysis any OHAs they indicated as unavailable [26,27].

**Accessibility:** Based on the EMHSLU 2016 and 2023, each medicine is expected to be available at certain public levels of care. We documented published survey results that checked the availability of medicines on the survey day at different levels of care in the public sector. In addition, we reviewed the overall availability of metformin 500 mg and glibenclamide 5 mg in public facilities as tracer drugs (vital medicines in Uganda) through the routine health facility reporting database [18,26].

**Affordability:** Since essential medicines are provided free of charge at public sector facilities, we focused instead on government affordability. We conducted a BIA following the guidelines of the International Society for Pharmacoeconomics and Outcomes Research (ISPOR) Task Force as an economic tool that assesses the cost of the adoption of the new interventions (see S2 Table for more details) [28]. To calculate the size of the population receiving each medicine, we reviewed the number of diabetic patients in each level of care in 2023 based on routine health facility data and the medicines that are required to be available at the respective level of care [18,26]. For each medication, we summed up the number of patients across all levels of care that are required to provide that medication. This information provides us with the total population that may potentially receive the drug. We then multiplied this total number by the estimated proportion of patients receiving each drug class to determine the target population for the BIA [29]. To calculate annual cost per patient, as shown in equation (1), we used 2023 medicines unit prices from the national warehouse catalogue-utilized WHO DDD as a reference for daily doses through the below formula. All costs are in local currency (UGX).

$$Annual\ Treatment\ Cost\ per\ Patient\ (UGX) = \frac{WHO\ DDD}{tablet\ strength} \times 365 \times Unit\ Price\ (UGX) \tag{1}$$

The final estimated total cost per medicine is ascertained by multiplying the total estimated number of diabetic patients receiving the medicine by the annual cost per patient for that medicine. Lastly, we converted the costs into USD according to the World Bank exchange rate in 2023 [15]. Finally, through two scenarios in the BIA, we calculated the potential cost-savings. The first scenario focuses on assessing substitutes within EMHSLU, while the second scenario evaluates other medicines available in the country but not in the EMHSLU (See S3 File for more details).

**Cost-Effectiveness:** We documented published cost-effectiveness studies on OHAs – identified through our literature review. Using World Bank health expenditure per capita data in the year, the results were published, and through the equation (2), we converted the Incremental Cost-Effectiveness Ratio (ICER) from different settings into an estimated equivalent ICER in Uganda [15].

$$Estimated\ ICER\ in\ Uganda\ (Int\$) = \frac{health\ expenditure\ per\ capita\ in\ Uganda\ \ (Int\$)}{health\ expenditure\ per\ capita\ in\ CEA\ setting\ \ (Int\$)} \times ICER\ in\ published\ study \tag{2}$$

In cases where the ICER was reported in currencies other than international US dollars (Int$), before calculating the estimated equivalent ICER for Uganda, we converted the ICER into Int$ using the World Bank official exchange rates for the respective year [15].

For the one CEA that allowed for a more precise adjustment of cost inputs, we used the formula mentioned above to adjust health spending and applied the Ugandan drug price.

**Clinical Profiles:** The safety and efficacy of available medicines within each OHA drug class in Uganda were appraised in accordance with published systematic reviews and meta-analyses covered in our literature review. We focused on universally reported safety and efficacy indicators, including risk of hypoglycemia, cardiovascular events, and all-cause mortality for safety; and HbA1c reduction for efficacy.

### EMHSLU enhancement

To determine recommended changes to EMHSLU Drug Selections (switches/additions/exclusions), we first narrowed down the OHAs options by short-listing the available and accessible medicines in Uganda. In the next step, we selected those medicines that are more cost-saving for the government through BIA results. Lastly, based on results from CEA and clinical effects studies, we proposed an alternative EMHSLU for consideration by the MoH in Uganda. We evaluated the results from characterizing EMHSLU to identify gaps related to any unsatisfactory outcomes with respect to the availability, accessibility, and affordability of OHAs in the list. Based on our findings on identified gaps and with respect to literature suggestions for overcoming each of the gaps, we proposed actions to enhance the efficacy of EMHSLU.

### Results

### Metrics for selection criteria

Metrics used in peer-reviewed papers for measuring availability, accessibility, and affordability of medicines were identified (See Table 1).

### Characterize EMHSLU

The background information about OHAs registered in Uganda, as well as a comparison between WHO EML and EMHSLU, can be found in Table 2. The major discrepancy between EMHSLU and the WHO EML is that EMHSLU includes medicines from the DPP-4 inhibitors and thiazolidinediones drug classes, which are not present in the WHO EML. Although glibenclamide 5 mg has been removed from the EMHSLU 2023, we included this medication in our

**Table 1. Summary of published metrics used for measuring availability, accessibility, and affordability.**

| Availability | |
|---|---|
| **Measure** | **Metric** |
| Medicine Registration [30] | List of medicines registered by the national FDA |
| Local Production and Import [31] | List of medicines produced locally or imported |
| Volume across time [32] | Volume of medicines available in the country |

| Accessibility | |
|---|---|
| **Measure** | **Metric** |
| Facility Surveys (WHO/HAI, Service Availability and Readiness Assessment, etc.) [33,34] | Medicine observed on survey day and/or stock-out check |
| Questionnaire (health workers/ patients) [35] | Health workers or patients report on medicine availability at facility levels |
| Supply chain system databases/ reports [36] | Routine data from health information systems |

| Affordability | |
|---|---|
| **Measure** | **Metric** |
| Catastrophic [37] | A threshold of "X% of income remaining after subsistence needs have been met". |
| Impoverishment [37] | The share of population whose income falls below poverty line after buying specific goods or services |
| WHO/HAI [37] | Total days' wages that the lowest-paid unskilled government worker need to work to buy a 30-day treatment regimen |
| Willingness to pay [38] | Willingness to pay $X for a good or service |
| Healthcare spending per capita [39] | Healthcare spending exceeding a certain percentage of household income (assessed based on monthly minimum wage, average monthly per capita income, monthly household net-adjusted disposable income, and monthly household financial wealth). |
| Income-based [40] | Time needed to earn enough to pay for a full treatment course out of pocket, based on average population wage |
| Affordability threshold [41] | The threshold of higher healthcare expenditure reduces spending on non-healthcare sectors |
| GDP per capita [23] | Percentage of GDP per capita allocated to healthcare costs |
| Budget Impact Analysis [42] | Financial consequences of adopting a new healthcare intervention |

analysis because it was listed in EMHSLU 2016 and is also included in the WHO EML. According to the MoH, glibenclamide 5 mg is still used in Uganda until the next fiscal year, pending a full transition to the new regimens [18].

According to EMHSLU, each essential medicine is required to be accessible at certain levels of care, in line with the cascade of care. This requirement means that patients who require more advanced regimens for T2DM management are expected to receive these medicines at higher levels of care (i.e., secondary and tertiary health facilities) (See Table 3). Currently, metformin 500 mg and glimepiride 2 mg are the only OHAs designated for availability in primary care settings. Compared to the EMHSLU 2016, the EMHSLU 2023 includes the addition of two types of sulfonylureas, gliclazide 80 mg and glimepiride 2 mg, while glibenclamide 5 mg has been excluded. Most of the newly added drugs, including gliclazide 80 mg, pioglitazone 30 mg, dapagliflozin 5 mg, and vildagliptin 50 mg, are designated for use only in tertiary healthcare facilities.

**Table 2. List of oral hypoglycemic agents registered in Uganda and comparison between WHO EML and EMHSLU.**

| Background | | | | |
|---|---|---|---|---|
| **Drug Class** | **Medicines Registered by Ugandan National Drug Authority [27]** | **EMHSLU (2016) [43]** | **EMHSLU (2023) [18]** | **WHO EML (2023) [44]** |
| Biguanides | Metformin 500 mg | Yes | Yes | Yes |
| | Metformin 750 mg | – | – | – |
| | Metformin 850 mg | – | – | – |
| | Metformin 1000 mg | – | – | – |
| Sulfonylureas | Glibenclamide 5 mg* | Yes | – | Yes |
| | Gliclazide 60 mg | – | – | Yes |
| | Gliclazide 80 mg | – | Yes | Yes |
| | Glimepiride 2 mg | – | Yes | – |
| | Glimepiride 3 mg | – | – | – |
| | Glimepiride 4 mg | – | – | – |
| | Glipizide 5 mg | – | – | – |
| Thiazolidinediones | Pioglitazone 30 mg | – | Yes | – |
| SGLT2 inhibitors | Dapagliflozin 5 mg | – | Yes | Yes |
| | Dapagliflozin 10 mg | – | – | Yes |
| DPP-4 inhibitors | Vildagliptin 50 mg | – | Yes | – |
| | Vildagliptin 100 mg | – | – | – |
| | Sitagliptin 25 mg | – | – | – |
| | Sitagliptin 50 mg | – | – | – |
| | Sitagliptin 100 mg | – | – | – |
| | Teneligliptin 20 mg | – | – | – |

**Notes:** The grey cells with "-" indicate that the medicines are not listed in the respective EMHSLU.

**Table 3. Oral Hypoglycemic Agents required by level of care and associated number of adult diabetic patients, Uganda 2023.**

| Level of Care | | Diabetes Medications Required at Each Care Level (Based on EMHSLU) | Number of Adult Diabetic Patients** |
|---|---|---|---|
| Primary Health Care | HC2 | – | 30,959 |
| | HC3 | Metformin 500 mg – Glimepiride 2 mg | 69,564 |
| Secondary Health Care | HC4 | Metformin 500 mg – Glibenclamide 5 mg* - Glimepiride 2 mg | 94,046 |
| | H | Metformin 500 mg – Glibenclamide 5 mg* - Glimepiride 2 mg | 87,763 |
| Tertiary Health Care | RR | Metformin 500 mg – Glibenclamide 5 mg* - Glimepiride 2 mg – Gliclazide 80 mg – Pioglitazone 30 mg – Dapagliflozin 5 mg – Vildagliptin 50 mg | 30,873 |
| | NR | Metformin 500 mg – Glibenclamide 5 mg* - Glimepiride 2 mg – Gliclazide 80 mg – Pioglitazone 30 mg – Dapagliflozin 5 mg – Vildagliptin 50 mg | 6,313 |

**Notes:** Abbreviations: HC2 = Health Center Level 2; HC3 = Health Center Level 3; HC4 = Health Center Level 4; H = Hospital; RR = Regional Referral Hospital; NR = National Referral Hospital. The gray cell with "-" indicates that there are no medicines designated to be available in the respective level of care.

*Glibenclamide 5 mg was in EMHSLU 2016 and removed from EMHSLU 2023.

**Based on Routine Health Facility Reporting Database.

**Availability (medication in Uganda).** Data on metformin 750 mg, gliclazide 60 mg, vildagliptin 100 mg, and sitagliptin 25 mg were unavailable from their suppliers. Therefore, we excluded these medicines from our analysis. As of 2024, there are two local pharmaceutical companies that produce OHAs in Uganda. Rene Industries Ltd makes metformin 500 mg and glibenclamide 5 mg, and Kampala Pharmaceutical Industries 1996 Limited makes metformin 500 mg [45,46]. Based on importation data from the MoH, India is identified as the source country for the majority of the generics of the OHAs listed in both EMHSLU 2016 and 2023 (See S1 Table for more details). Fig 1 illustrates the volume of imported medicines (total units) included in EMHSLU 2016 over a 2-year period from July 1, 2021, to June 30, 2023. Fig 2 presents the same information for medicines added to the EMHSLU in the 2023 version, covering the same time frame. Generally, medicines listed in the EMHSLU 2016 had higher import volumes compared to those added to the 2023 version. metformin 500 mg had the highest import volume, ranging from nearly 2 million to 16 million tablets imported each quarter, with no disruptions observed throughout the entire reviewed period. Conversely, dapagliflozin 5 mg, a newly added medicine to the EMHSLU, experienced the most import disruptions, being imported in only two out of the eight reviewed quarters. Gliclazide 80 mg, another recent addition to the EMHSLU, had the lowest import volumes, ranging from 2,800–37,800 tablets during the same period.

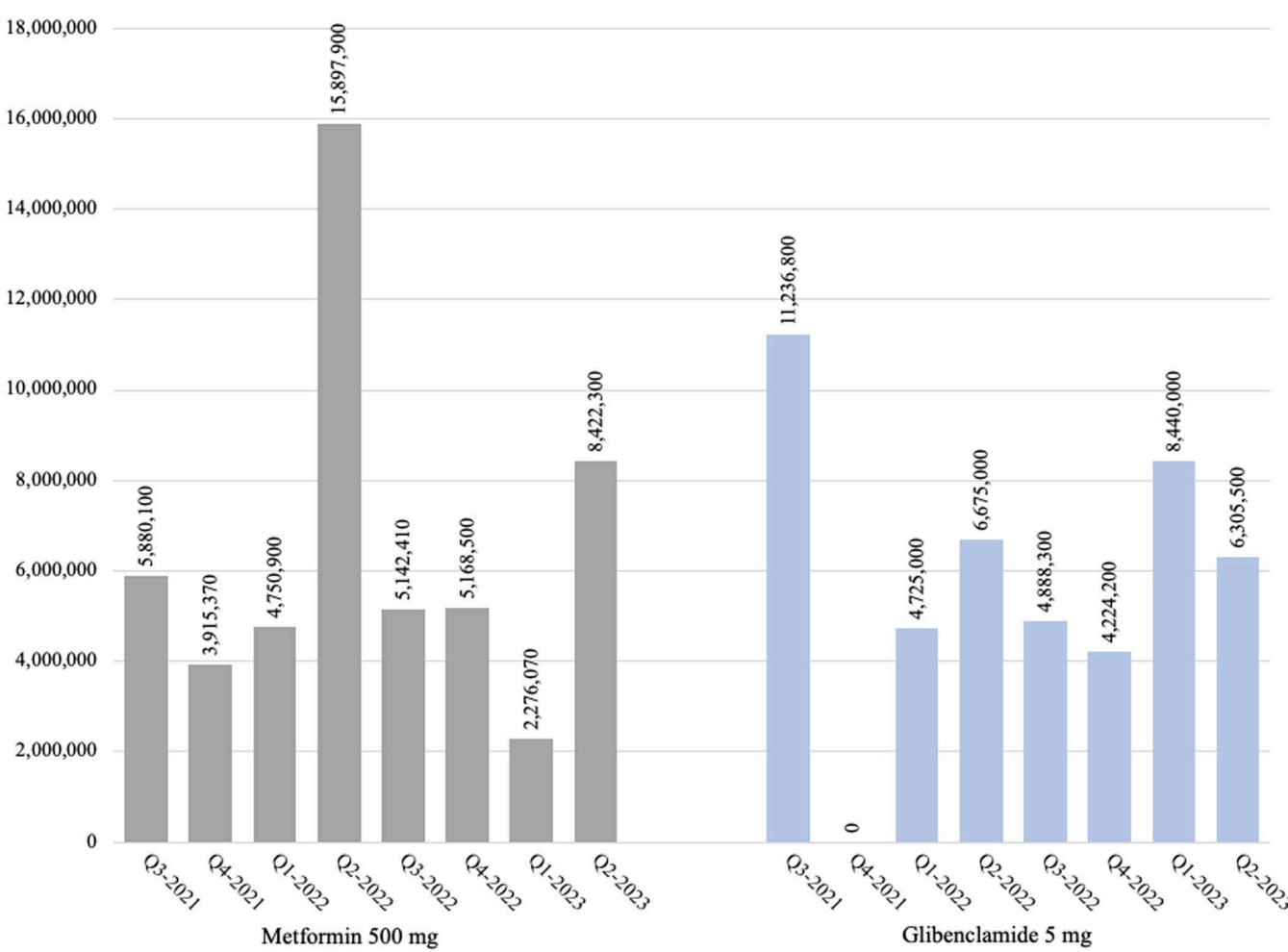

**Fig 1. The volume of imported oral hypoglycemic medicines which were in EMHSLU 2016 (Q3 2021–Q2 2023).**

  

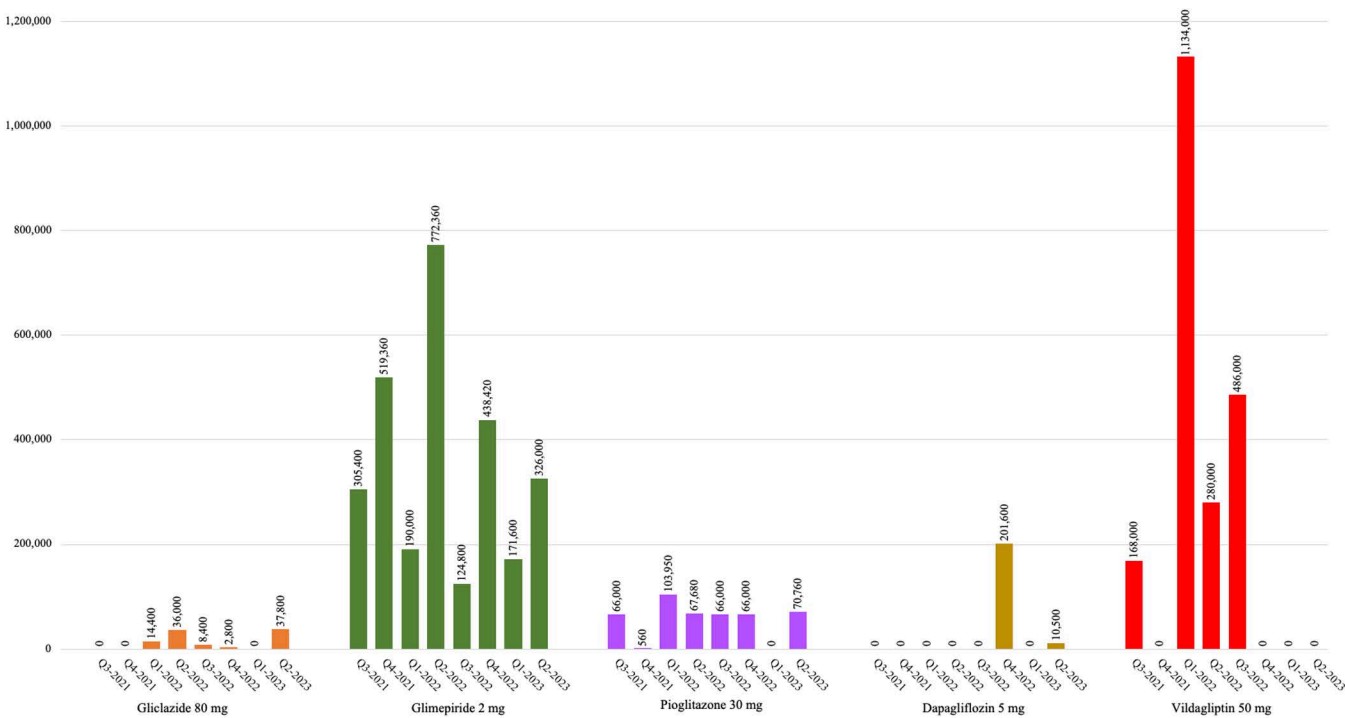

**Fig 2. The volume of imported oral hypoglycemic medicines added to the 2023 EMHSLU (Q3 2021–Q2 2023).**

**Accessibility (medicine in health facilities).** The routine health facility reporting database reveals that in 2023, metformin 500 mg was accessible at 62% of total facilities while glibenclamide 5 mg was accessible at 76% of facilities. Based on our review of published survey results, metformin 500 mg was accessible at more than 75% accessibility in Health Center 4 (HC4) and hospitals, whereas less than 25% of the Health Center 3 (HC3) facilities had this medicine. Similarly, glibenclamide 5 mg was accessible in more than 75% of the HC4 and hospitals that were assessed. There are no published surveys on those medicines that are added to the 2023 version of EMHSLU. However, according to the program and warehouse data, the accessibility of dapagliflozin 5 mg and sitagliptin 50 mg at the public facilities is zero due to budget limitations (See Table 4).

**Affordability (cost to government) and Budget Impact Analysis.** Our BIA examined potential savings with drug substitutions (See Table 5). There were no substitutes for pioglitazone 30 mg from the thiazolidinediones drug class available in Uganda. However, medicines from the other four drug classes had at least one substitute, either available in different doses or as comparable medicines within or outside the EMHSLU.

In Scenario 1, glibenclamide 5 mg was the least costly medicine among sulfonylureas in the EMHSLU, whereas gliclazide 80 mg was the costliest in this drug class. If glibenclamide 5 mg completely replaces glimepiride 2 mg and gliclazide 80 mg, it would yield potential savings of USD 2.65 million. In Scenario 2, for drugs in Uganda but outside the EMHSLU, replacing glimepiride 4 mg with glimepiride 2 mg and gliclazide 80 mg could save maximum USD 1.5 million and USD 600,000, respectively. Additionally, full replacement of dapagliflozin 5 mg with dapagliflozin 10 mg from the SGLT2 inhibitors class could save the government up to USD 230,000 (See S3 File for more details).

**Cost-Effectiveness.** We translated the evidence on the cost-effectiveness of OHAs in different settings into an estimated equivalent ICER for Uganda (See Table 6). Per WHO, an ICER of less than three times the GDP per capita is considered cost-effective [51]. We used health expenditure per capita to translate a published ICER in another setting into

Table 4. Accessibility of Oral Hypoglycemic Agents (OHAs) in the public sector in Uganda.

| Drug Class | OHAs Listed in EMHSLU 2016 and 2023 | Accessibility Status Based on the Published Survey Results* | | | | Accessibility Status Based on Routine Health Facility Reporting (2023)** |
|---|---|---|---|---|---|---|
| | | HC3 | HC3 | HC3 | HC3 | |
| Biguanides | Metformin 500 mg | 0 out of 2 (0%) [47] | 0 out of 2 (0%) [47] | 23 out of 26 (88.5%) [48] | 12 out of 13 (92.3%) [48] | 62% |
| | | | 27 out of 34 (79.4%) [49] | | | |
| | | 2 out of 12 (17%) [50] | 14 out of 14 (10%) [48] | | | |
| | | | 3 out of 4 (75%) [50] | | | |
| Sulfony-lureas | Glibenclamide 5 mg | – | 1 out of 2 (50%) [47] | 21 out of 26 (81%) [48] | 11 out of 13 (84.6%) [48] | 76% |
| | | | 26 out of 34 (76.5%) [49] | | | |
| | | | 11 out of 14 (78.6%) [48] | | | |
| | Gliclazide 80 mg | – | – | – | No Data | – |
| | Glimepiride 2 mg | – | – | No Data | No Data | – |
| Thiazoli-dinediones | Pioglitazone 30 mg | – | – | No Data | No Data | – |
| SGLT2 inhibitors | Dapagliflozin 5 mg | – | – | ~ 0 | ~ 0 | – |
| DPP-4 inhibitors | Vildagliptin 50 mg | – | – | ~ 0 | ~ 0 | – |

Notes: Abbreviations: HC3 = Health Center Level 3; HC4 = Health Center Level 4; H = Hospital; RR = Regional Referral Hospital; NR = National Referral Hospital. The grey cells with "-" indicate that the medicines are not required to be available at the respective level of care.

*Accessibility was measured as medicines available on the day of the survey.

**Accessibility status among all facilities expected to have the medicines.

the equivalent ICER in Uganda. Based on our calculations, all reviewed ICERs fall under three times the GDP per capita for Uganda and would be deemed cost-effective. Among the medicines not included in EMHSLU, sitagliptin has been shown to be cost-effective as an add-on to metformin compared to glibenclamide plus metformin. By adding sitagliptin to the EMHSLU, the MoH would spend between $2,342 and $3,066 per Quality-Adjusted Life Year (QALY), which is approximately equal to the country's GDP per capita at the time of the studies' publication. However, according to our BIA, vildagliptin, also a DPP-4 inhibitor, is cost-saving for the government. This pattern also aligns with a cost-effectiveness study in Greece, which found that vildagliptin plus metformin as a second-line treatment is dominant over glimepiride plus metformin in almost all reviewed scenarios.

As an example, for the study in Ecuador, we adjusted the pharmaceutical costs based on the unit prices in Uganda, and we deflated the prices to reach an estimation in 2021. Then, we adjusted other medical costs based on health expenditure per capita. The effectiveness in the original paper was mentioned as 0.02. Therefore, based on the new calculations, the ICER is $6,132 per QALY, while when adjusting all costs based on health expenditure per capita, the ICER was $2,324 per QALY. This significant difference underscores the challenge of geographically adopting an ICER due to the substantial variation in costs and utilization across different settings and highlights that further investigation is needed to accurately assess the cost-effectiveness of OHAs in Uganda.

**Clinical profiles.** Sulfonylurea and DPP-4 drug classes have more than one alternative medicine available in Uganda. In contrast, since only one medicine from each of the biguanides, thiazolidinediones, and SGLT2 inhibitors classes is

**Table 5. Budget impact analysis of oral hypoglycemic agents in Uganda.**

**Scenario 1 (potential changes using medicines within EMHSLU)**

| EMHSLU 2016 and 2023 | | | Cost-saving EMHSLU Alternatives for medicines in EMHSLU | |
|---|---|---|---|---|
| Drug Class | Medicines | Levels of care that need to have the medicines | Alternative | Maximum Cost-Saving (if fully replaced) |
| Biguanides | Metformin 500 mg | HC3, HC4, H, RR, NR | – | – |
| Sulfonylureas | Glibenclamide 5 mg | HC4, H, RR, NR | – | – |
| | Glimepiride 2 mg | HC3, HC4, H, RR, NR | Glibenclamide 5 mg | USD 2 million |
| | Gliclazide 80 mg | RR, NR | Glibenclamide 5 mg | USD 650,000 |
| | | | Glimepiride 2 mg | USD 400,000 |
| Thiazolidinediones | Pioglitazone 30 mg | RR, NR | – | – |
| SGLT2 inhibitors | Dapagliflozin 5 mg | RR, NR | – | – |
| DPP-4 inhibitors | Vildagliptin 50 mg | RR, NR | – | – |

**Scenario 2 (potential changes using medicines outside EMHSLU)**

| EMHSLU 2016 and 2023 | | | Cost-saving non-EMHSLU Alternatives for medicines in EMHSLU | |
|---|---|---|---|---|
| Drug Class | Medicines | Levels of Care that need to have the medicines | Alternative | Maximum Cost-Saving (if fully replaced) |
| Biguanides | Metformin 500 mg | HC3, HC4, H, RR, NR | – | – |
| Sulfonylureas | Glibenclamide 5 mg | HC4, H, RR, NR | – | – |
| | Glimepiride 2 mg | HC3, HC4, H, RR, NR | Glimepiride 4 mg | USD 1.5 million |
| | Gliclazide 80 mg | RR, NR | Glimepiride 4 mg | USD 600,000 |
| Thiazolidinediones | Pioglitazone 30 mg | RR, NR | – | – |
| SGLT2 inhibitors | Dapagliflozin 5 mg | RR, NR | Dapagliflozin 10 mg | USD 230,000 |
| DPP-4 inhibitors | Vildagliptin 50 mg | RR, NR | – | – |

**Notes:** Abbreviations: HC3 = Health Center Level 3; HC4 = Health Center Level 4; H = Hospital; RR = Regional Referral Hospital; NR = National Referral Hospital. The gray cells with "-" indicate that there is no cost-saving substitute for the respective medicine among the reviewed alternatives.

available in the country, a meaningful within-class comparison for these categories is not possible. The review of the safety and efficacy of available alternatives in each drug class is indicated in Table 7. Among the sulfonylureas, glipizide has the highest risk of adverse events, while the safety profile of gliclazide is more satisfactory. In terms of efficacy, glibenclamide lowers HbA1c more than other alternatives. Among the DPP-4 inhibitors, vildagliptin, sitagliptin, and teneligliptin have demonstrated high efficacy in managing T2DM with low risks of major adverse events. There were no significant differences in the efficacy and safety profiles among these drugs, making them comparable options for patients seeking DPP-4 inhibitor therapy.

## Discussion

The systematic approach used in this study to review the current medicines on the EMHSLU and develop recommendations—based on the WHO guideline for the selection of essential medicines at the country level—could serve as a useful model for other countries and disease areas. The concept of reviewing availability, accessibility, and affordability based on country contexts and incorporating budget impact analysis to select the most affordable medicines among cost-effective and clinically favorable options has been highlighted in various studies [3,68]. However, a comprehensive review that integrates all these factors to propose specific changes to a country's NEML has not been previously conducted. Through our analysis of availability, accessibility, and affordability and the review of published evidence on the cost-effectiveness and clinical effects of medicines, we identified areas for improvement in the provision of essential medicines

**Table 6.  Summary of cost-effectiveness analyses of oral hypoglycemic agents for type 2 diabetes mellitus.**

| Study | Setting | Perspective | Published Year | Intervention | Comparator | ICER in published study* | Health Expenditure per Capita Ratio** | Estimated Equivalent ICER in Uganda | 3 Times Uganda's GDP per Capita in the Same Year (Int$) |
|---|---|---|---|---|---|---|---|---|---|
| Cost-effectiveness of second-line antihyperglycemic therapy in patients with type 2 diabetes mellitus inadequately controlled on metformin [52] | Canada | Payer | 2011 | Metformin + Sulfonylurea | Metformin | $12,757 per QALY | 0.038 | $480 per QALY | 6815.19 |
| Cost-effectiveness study of oral hypoglycemic agents in the treatment of outpatients with type 2 diabetes attending a public primary care clinic in Mexico City [53] | Mexico | Society | 2012 | Glibenclamide | Metformin | $114.83 per QALY | 0.140 | $16 per QALY | 6107.25 |
| Pharmacoeconomic Analysis of Sitagliptin/Metformin for the Treatment of Type 2 Diabetes Mellitus: A Cost-Effectiveness Study [54] | Ecuador | Payer | 2021 | Sitagliptin + Metformin | Glibenclamide + Metformin | $19131.61 per QALY | 0.122 | $2342 per QALY | 8064.81 |
| Managing glycaemia in older people with type 2 diabetes: A retrospective, primary care-based cohort study, with economic assessment of patient outcomes [55] | UK | Payer | 2017 | Metformin + DPP-4 inhibitor | Metformin + sulfonylurea | £18,680 per QALY | 0.02 | $509 per QALY | 6481.98 |
| | | | | Metformin + DPP-4 inhibitor | Metformin + TZD | £15,343 per QALY | 0.02 | $418 per QALY | 6481.98 |
| Cost-Effectiveness of Saxagliptin Compared with Glibenclamide as a Second-Line Therapy Added to Metformin for Type 2 Diabetes Mellitus in Ethiopia [56] | Ethiopia | Healthcare System | 2021 | Saxagliptin + Metformin | Glibenclamide + Metformin | $2,259 per DALY averted | 1.455 | $3287 per DALY averted | 8064.81 |
| Costs and Consequences Associated with Newer Medications for Glycemic Control in Type 2 Diabetes [57] | USA | Healthcare System | 2010 | Sitagliptin + Metformin | Glibenclamide + Metformin | $169,572 per QALY | 0.018 | $3066 per QALY | 6284.19 |
| Cost effectiveness of vildagliptin versus glimepiride as add-on treatment to metformin for the treatment of diabetes mellitus type 2 patients in Greece [58] | Greece | Social Insurance Funds | 2017 | Vildagliptin + Metformin | Glimepiride + Metformin | Dominant in most scenarios. | 0.038 | Dominant in most scenarios. | 6481.98 |

Notes:

*ICERs in currencies other than US$ are converted to US$ for calculation.

**Healthcare spending per capita (Int$, based on PPP) in Uganda in the year the results published/ Healthcare spending per capita (Intl$) in the country reviewed in the year the results published.

for the management and control of T2DM in Uganda. In addition to the important steps that the government could take to enhance the availability, accessibility, and affordability of OHAs listed in the EMHSLU 2023, our BIA suggests potential OHA alternatives that can be added or replaced in the EMHSLU, offering significant cost-savings while meeting cost-effectiveness and clinical effects criteria.

### Recommended changes to EMHSLU drug selections (switches/ additions/ exclusions)

Based on our BIA and review of cost-effectiveness analysis and clinical profiles of medicines, we propose adding glimepiride 4 mg and dapagliflozin 10 mg to the EMHSLU, in addition to retaining their lower doses on the EMHSLU to support

**Table 7. Clinical profiles (safety and efficacy) of sulfonylureas and DPP-4 inhibitors in type 2 diabetes management.**

| Drug Class | Medicines | Risk of Adverse Events | | | Blood Glucose Level Control |
|---|---|---|---|---|---|
| | | Hypoglycemia | Cardiovascular Events | All-Cause Mortality | HbA1c Reduction |
| Sulfonylureas | Glibenclamide | High* [59] | Low [60] | Low [60] | High [61] |
| | Gliclazide | Low [59] | Low [60] | Low [60] | Moderate [61] |
| | Glimepiride | Moderate [59] | Moderate [60] | Low [60] | Moderate [61] |
| | Glipizide | Moderate [59] | Moderate [60] | High [60] | Moderate [61] |
| DPP-4 inhibitors | Vildagliptin | Low [62] | Low [62] | Low [62] | High [63] |
| | Sitagliptin | Low [64] | Low [64] | Low [65] | High [63] |
| | Teneligliptin | Low [66] | Low [67] | Low [67] | High [66] |

Notes:

*especially in patients over 65 years old.

better compliance for patients who need the lower doses [69]. We also suggest excluding gliclazide 80 mg due to its high cost and lack of significant clinical benefits compared to glimepiride 2 mg and 4 mg. Although glibenclamide 5 mg has been removed from the EMHSLU 2023, per our review and the WHO EML, we recommend keeping it in the system for patients under 65 years old to avoid severe hypoglycemia in elderly people as it is the most cost-saving medicine among the sulfonylureas with a great efficacy profile [44]. For patients over 65 years old, glimepiride is a suitable alternative with satisfactory clinical effects in addition to its cost-savings. (See Table 8 for more details).

## Programmatic/managerial challenges for EMHSLU

The below actions are suggested to enhance the availability, accessibility, and affordability of OHAs in EMHSLU:

**Availability.** While local production of metformin 500 mg and glibenclamide 5 mg has supplemented OHAs imports in Uganda, for medicines that were newly added to the EMHSLU, the import trend by mid-2023 had been neither sufficient nor consistent to address the demand of the estimated population requiring these drugs. Although lower availability of medicines newly added to the EMHSLU is expected during the transition period, strengthening the functions of NRAs, such as registration and market authorization, licensing establishments, and market surveillance and control, will enhance the uninterrupted availability of medicines and ensure high-quality medicines are available in the country. Moreover, expanding local production of medicines is crucial for maintaining a steady supply and improving access [70,71].

**Accessibility.** As the new EMHSLU was revised in 2023, there are no published surveys yet on the accessibility of the newly added medicines. However, our results suggest that budget limitations have significantly impacted the accessibility of medicines, such as dapagliflozin 5 mg and vildagliptin 50 mg. Since the supply chain system plays a crucial role in ensuring the geographical accessibility of medicines, effective supply chain management, including accurate forecasting, procurement of necessary dosage forms, timely delivery to health facilities, and appropriate storage, needs to be strengthened in Uganda for improved functionality [72,73]. Additionally, frequent stock-outs of essential medicines in public health facilities often force patients to turn to private clinics and pharmacies to continue their treatment which can compromise affordability and continuity of care [74].

**Affordability.** The MoH in Uganda allocates UGX 435 billion (13% of its total budget) to the treatment and control of DM, with OHAs consuming a significant portion of this expenditure. To manage costs and improve access, the government has engaged in price negotiations and signed Memoranda of Understanding (MoUs) with pharmaceutical companies like Novartis. This agreement, although it expired in 2023 and has not yet been renewed, includes the procurement of 15 NCD medications, such as vildagliptin, at the cost of $1 for a one-month treatment course. Expanding such MoUs with pharmaceutical companies for other OHAs can significantly enhance affordability from the government's perspective as the buyer.

**Table 8. Recommended changes to EMHSLU drug selection for oral hypoglycemic agents.**

| Drug Class | Medicines Registered by Ugandan National Drug Authority | EMHSLU (2016) | EMHSLU (2023) | Recommended Changes | Justification for the Recommended Changes |
|---|---|---|---|---|---|
| Biguanides | Metformin 500 mg | Yes | Yes | Keep | Cost-Saving |
| | Metformin 850 mg | – | – | Do not Add | Not Cost-Saving |
| | Metformin 1000 mg | – | – | Do not Add | Not Cost-Saving |
| Sulfony-lureas | Glibenclamide 5 mg | Yes | – | Keep | Cost-Saving and Clinically Favorable for Patients Under 65 |
| | Gliclazide 80 mg | – | Yes | Remove | Not Cost-Saving |
| | Glimepiride 2 mg | – | Yes | Keep | Better Patient Compliance with Lower-Dose Option |
| | Glimepiride 4 mg | – | – | Add | Cost-Saving |
| | Glipizide 5 mg | – | – | Do not Add | Neither Cost-Saving nor Clinically Favorable |
| Thiazoli-dinediones | Pioglitazone 30 mg | – | Yes | Keep | Only Available Medicine from This Drug Class in Uganda |
| SGLT2 inhibitors | Dapagliflozin 5 mg | – | Yes | Keep | Better Patient Compliance with Lower-Dose Option |
| | Dapagliflozin 10 mg | – | – | Add | Cost-Saving |
| DPP-4 inhibitors | Vildagliptin 50 mg | – | Yes | Keep | Cost-Saving |
| | Sitagliptin 50 mg | – | – | Do not Add | Not Cost-Saving |
| | Sitagliptin 100 mg | – | – | Do not Add | Not Cost-Saving |
| | Teneligliptin 20 mg | – | – | Do not Add | Not Cost-Saving |

**Notes:** The grey cells with "-" indicate that the medicines are not listed in the respective EMHSLU.

In addition to the above-mentioned areas for improvement, according to WHO guidelines for the selection of essential medicines, clinical guidelines should play a vital role in this process [1]. In Uganda, though, the 2023 national clinical guideline lacks some of the OHAs listed in the EMHSLU 2023, mentioning only metformin, glibenclamide, and glimepiride [75]. Aligning the national clinical guidelines with the NEML is necessary to ensure that the health workforce adheres to standardized treatment protocols and drug choices [1]. This would enhance NEML efficiency and ultimately greater cost-savings by ensuring that patients receive appropriate medicines according to their clinical profiles [76].

## Limitations

We reviewed published cost-effectiveness analyses on OHAs through our literature review; however, systematic reviews and meta-analyses are needed for a better understanding of what medicines within each drug class are more cost-effective. Additionally, we translated ICERs from other settings into an estimated ICER for Uganda, using the simplifying assumption that there is a link between ICERs and health expenditure per capita, whereas a detailed analysis for cost and utility adjustment is required to precisely adapt the ICERs geographically. This study also relied on summarized clinical profiles reported in existing systematic reviews and meta-analyses. A more rigorous review of reviews may be needed to capture the full range of safety and efficacy outcomes for oral hypoglycemic agents and produce more granular estimates tailored to the Ugandan context.

We used WHO DDD for our BIA as a reference for daily defined doses, and we utilized published literature results to review the share of each drug class in the T2DM patients' prescriptions. However, detailed consumption data is required to calculate the BIA accurately especially as this BIA is focused on procurement-level costs and did not account for additional supply chain considerations such as inventory holding costs, packaging-related storage differences, etc. Moreover,

it should be highlighted that these results are based on current unit prices, which may not remain stable due to inflation or future price negotiations with companies. Lastly, this analysis did not account for the potential downstream costs associated with adverse clinical outcomes, such as hypoglycemia, which may be more common with certain medicines like glibenclamide and could impact some of the projected cost savings. We propose considering BIA as one of the main decision-making tools in the process of revising the EMHSLU in Uganda.

While our primary focus for availability was on the distinction between local and global production, other factors—such as global demand for certain medicines, the proximity of production sources, and regulatory compliance of manufacturers—may also influence the sustainability of supply chains for imported medicines [32,77]. We also were unable to assess how changes in demand resulting from substitutions in the EMHSLU might influence medicine availability or accessibility in Uganda. Future studies should examine NEMLs that have associated demand data to further inform the optimization of NEMLs.

## Conclusion

The government of Uganda has committed to moving towards achieving UHC and has put forth efforts to develop national health insurance, expand public health facilities, and enhance health services [78]. Despite the costs, many patients diagnosed with T2DM prefer private-for-profit clinics over public healthcare facilities because these clinics often have better availability of medicines [21]. This trend, however, compromises affordability for patients since medicines in the private sector are not free [79].

The EMHSLU has several strengths, including a strong commitment from Uganda's Ministry of Health to regularly revise and update the list based on population needs, health workforce capacity, and the quality and safety of medicines [18]. However, our analysis reveals some gaps in the availability, accessibility, and affordability of essential medicines for managing T2DM in Uganda. While local production and MoUs with pharmaceutical companies have helped, the inconsistent supply of newly added medicines and budget constraints continue to limit access. Our BIA suggests including glimepiride 4 mg and dapagliflozin 10 mg in the EMHSLU while keeping glibenclamide 5 mg, which also enhances cost-effectiveness and clinical outcomes. Additionally, aligning national clinical guidelines with the NEML is essential to ensure standardized treatment protocols and improve patient care.

## Supporting information

**S1 Table. Source countries of imported Oral Hypoglycemic Agents (OHAs) in Uganda.** Checklist used to guide and assess the completeness of the budget impact analysis based on ISPOR guidelines.
(DOCX)

**S2 Table. Budget Impact Analysis Checklist (ISPOR).** Checklist used to guide and assess the completeness of the budget impact analysis based on ISPOR guidelines.
(DOCX)

**S3 File. Budget Impact Analysis.** Excel file containing detailed calculations and model inputs used in the budget impact analysis.
(XLSX)

## Author contributions

**Conceptualization:** Atousa Bonyani, Tracy K. Lin, Ambrose Jakira, James G. Kahn.

**Data curation:** Atousa Bonyani, Tracy K. Lin, Ambrose Jakira, Isaac D. Kimera, Martin Muddu, Jeremy I. Schwartz, James G. Kahn.

**Formal analysis:** Atousa Bonyani, Tracy K. Lin, James G. Kahn.

**Methodology:** Atousa Bonyani, Tracy K. Lin, James G. Kahn.

**Project administration:** Atousa Bonyani, Ambrose Jakira, Isaac D. Kimera, Martin Muddu.

**Resources:** Ambrose Jakira, Isaac D. Kimera, Martin Muddu, Jeremy I. Schwartz.

**Supervision:** Tracy K. Lin, Ambrose Jakira, Jeremy I. Schwartz, James G. Kahn.

**Validation:** Ambrose Jakira, Isaac D. Kimera, Martin Muddu, Jeremy I. Schwartz.

**Visualization:** Atousa Bonyani, Tracy K. Lin, James G. Kahn.

**Writing – original draft:** Atousa Bonyani, Tracy K. Lin, James G. Kahn.

**Writing – review & editing:** Atousa Bonyani, Tracy K. Lin, Ambrose Jakira, Isaac D. Kimera, Martin Muddu, Jeremy I. Schwartz, James G. Kahn.

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
