## [Decision Letter · Decision Letter 0]

Dear Dr. Bonyani,

Thank you for submitting your manuscript to PLOS ONE. After careful consideration, we feel that it has merit but does not fully meet PLOS ONE’s publication criteria as it currently stands. Therefore, we invite you to submit a revised version of the manuscript that addresses the points raised during the review process.

**ACADEMIC EDITOR:**

We look forward to receiving your revised manuscript.  

Kind regards,

Sonak D. Pastakia

Academic Editor

PLOS ONE

Journal Requirements:

2. Please amend either the title on the online submission form (via Edit Submission) or the title in the manuscript so that they are identical.

Additional Editor Comments:

The authors present a novel and interesting analysis of the unique cost considerations faced by LMIC governments when considering which medications should be added to their national formulary. While many take a simplistic view on formulary decisions and assume that the newer medications are simply too expensive to provide within public sector health systems, the authors provide a much more nuanced evaluation that considers the multiple different factors which governments should look at when making these decisions. The authors provide a framework for an economic analysis/budget impact analysis that would be of considerable benefit to additional LMIC’s facing similar decisions regarding the management of decisions.

Page 15 line 282 -what does health center 4 mean?

“Similarly, glibenclamide 5 mg was accessible in 282 more than 75% of the health center 4 and hospitals that were assessed” (minor)

In the discussion of sulfonylureas in the affordability section on page 17, the difference in cost of the different medications is discussed but the clinical differences are not incorporated leaving this assessment incomplete. For example, the major advantage of gliclazide is the enhanced safety amongst patients with renal insufficiency and decreased risk of hypoglycemia as described in table 7. From a health economics perspective, the number of patients not experiencing severe hypoglycemia should be factored into the comparative assessment as opposed to treating all of these medications as equivalent in terms of cost. (moderate)

In table 7, why aren't biguanides or SGLT2i's included as well?  (minor)

Is there a reason GLP-1's are mentioned anywhere? (minor)

Reviewers' comments:

Reviewer's Responses to Questions

**Comments to the Author**

1. Is the manuscript technically sound, and do the data support the conclusions?

Reviewer #1: Yes

Reviewer #2: Yes

2. Has the statistical analysis been performed appropriately and rigorously?

Reviewer #1: No

Reviewer #2: Yes

3. Have the authors made all data underlying the findings in their manuscript fully available?

Reviewer #1: Yes

Reviewer #2: Yes

4. Is the manuscript presented in an intelligible fashion and written in standard English?

Reviewer #1: Yes

Reviewer #2: Yes

Reviewer #1: 1. Summary

The paper focuses on identifying the list of medicines that should be included in a country’s Essential Medicines and Health Supplies List (EMHSL), with the ultimate goal of improving cost-savings and health outcomes. This is done through a case study of five classes of Oral Hypoglycemic Agents (OHAs) for type 2 diabetes mellitus in the context of Uganda: biguanides, sulfonylureas, thiazolidinediones, SGLT-2 inhibitors, and DPP-4 inhibitors. The paper evaluates these medicines based on criteria such as availability, accessibility, government affordability, cost-effectiveness, safety, and efficacy. Government affordability is then assessed under two different scenarios: (i) substituting products within the EMHSL; and (ii) incorporating products outside the EMHSL. The findings show that under scenario 1, a complete replacement of glimepiride 2 mg and gliclazide 80 mg with glibenclamide 5 mg could save the Ugandan government $2.65 million per year. Similarly, the replacement of dapagliflozin 5 mg and glimepiride 2 mg with their fuller doses (i.e., dapagliflozin 10 mg and glimepiride 4 mg) would save the Ugandan government up to $0.60 million per year.

The paper addresses a critically important topic with significant implications for public health: designing the list of health supplies that are to be offered on the EMHSL. If the recommendations were to be taken up and implemented by policymakers, they could impact the health landscape for an entire country. Given the high stakes involved, I believe the methodology and the resulting recommendations should be strengthened to improve their scientific rigor and validity. I elaborate on these concerns and offer a few recommendations below.

2. Major Areas of Improvement

2.1. Methodology

- On page 9 and in Table 3, the paper calculates the size of the population receiving each medicine based on “routine health facility data and the medicines that are required to be available at the respective level of care.” Please provide more detailed information on the calculations and assumptions that you used to arrive at the numbers in Table 3. If you used point-of-care dispensation data at point-of-care clinics, I’m concerned that your numbers likely to be right-censored (due to products being out of stock, etc.) Hence, your estimates would be significantly underestimating the total number of patients with diabetes in the country.  (Moderate-I would like to see some comment on how you address the underestimation of patients as a result of stockouts)

For instance, the data presented in Table 3 suggests that the total population of adult diabetic patients in Uganda is approximately 320,000 across all medications and levels of care. However, based on the 4.6% prevalence of diabetes in Uganda, as noted on page 27, and a total population of around 50 million people, the estimated number of diabetic patients should be closer to 2,300,000. This discrepancy indicates that your calculations may underestimate the actual patient population. Please clarify this inconsistency, as it raises concerns about the accuracy and reliability of the findings. (Moderate-I would include a comment on this discrepancy in the paper while still acknowledging that countries like Uganda have the highest rate of underdiagnosed diabetes and struggle with inconsistent follow-up)

- Similarly, the percentage last-mile availability of health products based on “routine health facility reporting” data—reported as 62% and 76%—are likely to be biased. For example, the 62% last-mile availability of Metformin 500 mg could be due to a variety of different factors such as high popularity among the population leading to stock-outs, low popularity leading health facilities to not prioritize the product, or lack of sufficient procurement quantities at the national level. With that in mind, an alternative to completely replacing a drug on the EMHSL could involve improving last-mile availability or increasing procurement quantities. How do these alternative interventions compare in your context, and what impact might they have on accessibility, affordability, and health outcomes? (Moderate)

- Please provide detailed information on where your estimated cost savings originate from. Are these purely based on the difference between the average cost of procuring the original health products and your recommended substitutes? If so, there are a few additional factors that may need to be taken into account during the cost calculation process. For example, how comparable are the inventory holding costs for the two types of products given potential differences in packaging size (e.g., Glimepiride 4mg vs. Gliclazide 80 mg)? How about obsolescence and expiration related costs? (Major-more description of these estimations are important to help the reader understand your calculations)

- In Table 6, the paper utilizes data from countries other than Uganda in order to evaluate the cost-effectiveness of various drugs in the Ugandan context. Some of the countries listed, however, exhibit very different health infrastructure, costs, and health insurance models than Uganda (e.g., Canada, US, UK). Please clarify if and why these are valid benchmarks for Uganda. (Minor-please comment on the inherent limitations with the lack of data from countries more similar to Uganda)

- Table 7 provides a summary of the clinical profiles (safety and efficacy) across different classes and types of products. This is a good starting point, but I believe for the results of the study to be valid, this analysis needs to be conducted more rigorously. For example, would it be possible to develop more precise measurements of safety and efficacy beyond the current categorical classification of low, moderate, and high? Further, please clarify whether or not the safety and efficacy metrics in Table 7 are universal clinical profiles that are likely to hold in various settings. Or do these need to be modified depending on the specific characteristics of each country (e.g., based on demographic factors, disease prevalence, comorbidities, etc.)?  (Major)

2.2. Terminologies

- The terminologies used in the paper need to be defined more clearly, more upfront, and used consistently throughout the manuscript. For example, “availability” is defined at the national level, where as “accessibility” is also referring to availability but at the more granular and last-mile health facility level. This is somewhat confusing. Should these be referred to as national availability and last-mile availability instead? (Moderate)

- Similarly, the definition of “affordability” in the paper is rather unorthodox, being defined as the affordability of a health product with respect to government. However, affordability is most commonly defined—in both academia and practice—as products being affordable to patients and clients who are end-users (and not government). In this context, cost-effectiveness seems like a more appropriate terminology.  (Minor-I understand that medications are provided for free in the public sector making defining this term slightly problematic)

- Quite confusingly, the paper has a separate category named “cost-effectiveness” which is defined as “the costs and health outcomes associated with pharmaceutical interventions, comparing the relative expenses and benefits of different medications to determine.” How is this differentiated from “affordability”?  (Major)

- On page 7, the paper defines “affordability” in three different ways: ”the product value dimension considers cost-effectiveness and Quality Adjusted Life Year (QALY), the healthcare system perspective dimension focuses on optimizing the health system budget using budget impact analysis (BIA), and the family perspective dimension considers minimum worker wages or Gross Domestic Product (GDP).“ These are very different definitions with substantially different implications. Which of these three is the definition adopted by the paper? It seems like it’s government affordability. Again, there are a lot of inconsistencies in definitions that are confusing to the reader. (Major)

3. Minor Areas of Improvement

- The background and motivation for the paper is rather weak. Please provide more detailed information on why this topic is important. For example, what is the process for assigning a health product to the EMHSL? What are the implications when a product is added to the list and when it is taken off? What is an essential medicine or health product? (Minor - I found the background and motivation for this paper to be quite strong as many LMIC governments would benefit from this more nuanced discussion of cost-effectiveness)

- Please provide some justification as why Uganda was selected as the case study. Why diabetes? This is valuable information that should not be left out. Further, how can your results be useful for other entities beyond the government of Uganda (e.g., other national governments, NGOs, etc.)?  (Minor- I assume the investigators have ongoing collaborations in Uganda facilitating the availability of data for this analysis)

- In calculating the national supply availability, local vs. global production is a relevant factor that the paper has highlighted. Beyond that, I believe the proximity of global production supply base is another relevant factor that could be taken into consideration (this could determine the risk of potential disruptions). (minor)

In sum, it was a pleasure reviewing this paper and I hope the authors find my comments helpful in further advancing this research. Best of luck with the next steps!

Reviewer #2: Thank you for the opportunity to review the paper "Economic Assessment of Potential Changes to Essential Medicines for Diabetes in Uganda". This research presents an innovative and timely study that provides valuable insights into improving access to essential medicines in LMICs. The approach adopted in this research, addresses critical issues related to both health outcomes and cost-savings, making a strong contribution to the ongoing discussion of optimal medication policies.

The proposed alternatives to current medications in the EMHSLU are not only grounded in evidence. The interdisciplinary approach of the research team, including members from Uganda, enhances the credibility and contextual relevance of the study. Moreover, the combination of a thorough literature review, data gathered directly from the Ministry of Health (MoH), and the BIA strengthens the overall argument for the proposed changes. Similarly, the identification of the limitations of translating the Cost-Effectiveness Analysis (CEA) studies into the Ugandan context, adds valuable transparency to the research and highlight the importance to work towards more comparable metrics among different contexts.

I would like to offer a few suggestions for improving the manuscript:

Distribution Challenges: The paper mentions that the inclusion of newer OHAs in the EMHSLU has led to issues of availability and accessibility, comparing them to older drugs like metformin or glibenclamide, which have a long history of use. While these challenges are significant, I believe it would be beneficial to emphasize the transitional nature of these issues. Strengthening and adapting the medication supply chain for newer drugs should be viewed as an expected part of the process, rather than framing it as a disadvantage of the new medications themselves. (moderate)

Glibenclamide Withdrawal: The MoH’s decision to remove glibenclamide from the essential medicines list, in line with current clinical practice guidelines due to safety concerns, is a key point. It would be beneficial to explore if the potential costs associated with treating complications like hypoglycemia, which could counterbalance the savings identified in the study.  (Major- this is an important consideration that overlaps with the comments about the difference between gliclazide and other sulfonylureas)

The authors describe that the availability of medications varies depending on the complexity of the hospital network. It would be interesting to explore how this is related to the clinical profiles of patients. Additionally, a more in-depth discussion on how clinical profiles and the clinical practice guidelines' recommendations could lead to greater efficiencies would be valuable. This could have an impact on the BIA calculations by better segmenting the patient population, ultimately generating higher cost savings. (Minor)

Epidemiological Context: It would be helpful to introduce data on the burden of type 2 diabetes in Uganda earlier in the paper provide context for the study and its significance. (minor)

Repetition of Recommendations: Finally, the section "recommended changes to EMHSLU drug selections" repeats the recommendation to retain both doses of glimepiride and dapagliflozin (minor)

**Do you want your identity to be public for this peer review?** For information about this choice, including consent withdrawal, please see our Privacy Policy

Reviewer #1: No

Reviewer #2: No

---

## [Author Response · Author response to Decision Letter 1]

5 May 2025

Please see the attached Response to Reviewers document for detailed, point-by-point replies to all reviewer and editor comments.

---

## [Decision Letter · Decision Letter 1]

Dear Dr. Bonyani,

Thank you for submitting your manuscript to PLOS ONE. After careful consideration, we feel that it has merit but does not fully meet PLOS ONE’s publication criteria as it currently stands. Therefore, we invite you to submit a revised version of the manuscript that addresses the points raised during the review process.

**The paper is largely acceptable and would like you to consider a few of the recommendations from Reviewer 1.  Please provide a quick response to those comments and we can proceed with providing a full acceptance**

We look forward to receiving your revised manuscript.

Kind regards,

Sonak D. Pastakia

Academic Editor

PLOS ONE

Journal Requirements:

Additional Editor Comments:

The paper is much improved as a result of your revisions and responses to the first set of reviews. There are a couple of minor suggestions to consider but the paper is largely acceptable for publication.  Please review the comments and the commentary from me in parentheses.   The next review will be quick.   

Reviewers' comments:

Reviewer's Responses to Questions

**Comments to the Author**

Reviewer #1: All comments have been addressed

Reviewer #2: (No Response)

2. Is the manuscript technically sound, and do the data support the conclusions?

Reviewer #1: Yes

Reviewer #2: No

3. Has the statistical analysis been performed appropriately and rigorously?

Reviewer #1: Yes

Reviewer #2: No

4. Have the authors made all data underlying the findings in their manuscript fully available?

Reviewer #1: Yes

Reviewer #2: Yes

5. Is the manuscript presented in an intelligible fashion and written in standard English?

Reviewer #1: Yes

Reviewer #2: Yes

Reviewer #1: I read the revised version of the paper carefully and with great interest. I applaud the authors’ concerted efforts in addressing the review team’s comments in the previous round. The paper is now stronger from both a methodological and expositional standpoint. My comments in this round are aimed at making a few additional incremental improvements, as highlighted below.

− On page 10, please correct the typo by changing the denominator from ”tablet strenght” to “tablet strength.” (minor, please correct)

− Please number all equations accordingly. (minor, please correct)

− In Table 8, the recommended change for “Glibenclamide 5 mg” is to “Keep” the drug. However, this drug was already taken off the essential medicine list in 2023. Would it not make more sense if the recommendation is to instead “Add” this drug? (minor, please explain)

− Would it be possible to create a table similar to table 8 but with summary information on the (i) justifications for the recommended changes (e.g., cost savings, lack of clinical benefits, etc.) along with the (ii) pragmatic and managerial challenges for implementing these recommendations (e.g., lack of local production capacity, etc.)? That would provide the reader with a concise overarching picture of all they need to know regarding the paper’s actionable implications. (minor, please consider as this would greatly improve clarity)

Thanks again for the opportunity to review this paper and wish you the best of luck on the next steps!

Reviewer #2: Thank you for the opportunity to review this manuscript. I appreciate the authors’ efforts in addressing an important public health issue and proposing a structured methodology that incorporates more variables than traditional HTA processes, including last-mile accessibility.

However, there are significant limitations in the analysis, particularly regarding the comparability of associated costs. A more robust model would benefit from including additional cost-related factors, such as the dynamics of medication adoption during transition periods, scenario analyses to estimate future uptake of drug classes, and inventory holding costs, all of which could refine the budget impact estimates. Moreover, while adverse effects are more frequent and severe in individuals over 65, a group that may represent a small proportion of Uganda’s population (though this is not specified), these events are closely linked to treatment adherence. Poor adherence increases the risk of both short- and long-term complications, so projected savings should be adjusted to reflect the potential costs of adverse events and non-adherence.  (These comments can be considered in future efforts, however, I don't believe that they negate the value of the current initial attempt)

The evidence currently presented may not be sufficient to fully support the conclusions drawn. The authors state that the objective of the study is to "recommend changes to the EMHSLU drug selections based primarily on reduced costs, as well as cost-effectiveness and medicines’ clinical profiles." Yet, the recommendations seem to be based primarily on the direct costs of medicines. Since this is proposed as a novel approach to inform changes to National Essential Medicines Lists (NEMLs) in low- and middle-income countries (LMICs), it would be valuable to include a more comprehensive model that integrates the additional variables identified by the authors and those emphasized by WHO.   (These comments can be considered in future efforts, however, I don't believe that they negate the value of the current initial attempt)

While the manuscript discusses several important factors, these are assessed individually rather than within a unified, comparative framework. An integrated approach that quantifies these elements economically within a replicable model would offer a stronger basis for policy-making. Such a model should reflect the chronic nature of type 2 diabetes, allowing for projections across multiple cycles and capturing both the consequences of listing medicines that are not effectively accessible and the trade-offs associated with using more accessible, lower-cost alternatives that may carry a higher risk of adverse effects. Incorporating these dynamics would improve the relevance and applicability of the analysis for real-world decision-making in LMICs.   (These comments can be considered in future efforts, however, I don't believe that they negate the value of the current initial attempt)

**Do you want your identity to be public for this peer review?** For information about this choice, including consent withdrawal, please see our Privacy Policy

Reviewer #1: No

Reviewer #2: No

---

## [Author Response · Author response to Decision Letter 2]

4 Jun 2025

Comment 1: On page 10, please correct the typo by changing the denominator from “tablet strenght” to “tablet strength.” (minor, please correct)

Response: We thank the reviewer’s comment. The typo is fixed.

Comment 2: Please number all equations accordingly. (minor, please correct)

Response: Thank you for this valid comment. We have added the equation numbers to the manuscript.

Comment 3: In Table 8, the recommended change for “Glibenclamide 5 mg” is to “Keep” the drug. However, this drug was already taken off the essential medicine list in 2023. Would it not make more sense if the recommendation is to instead “Add” this drug? (minor, please explain)

Response: We appreciate this important point. Since the medicine is still in the health system and the government is in the transition phase to remove glibenclamide, we used the word “Keep” instead of “Add”.

Comment 4: Would it be possible to create a table similar to table 8 but with summary information on the (i) justifications for the recommended changes (e.g., cost savings, lack of clinical benefits, etc.) along with the (ii) pragmatic and managerial challenges for implementing these recommendations (e.g., lack of local production capacity, etc.)? That would provide the reader with a concise overarching picture of all they need to know regarding the paper’s actionable implications. (minor, please consider as this would greatly improve clarity)

Response: We truly appreciate the reviewer’s thoughtful comment. We have added a new column to Table 8 titled “Justification for the Recommended Changes.” Regarding the pragmatic and managerial challenges, we believe that presenting this information in the text is more helpful to readers, as a table may not add additional clarity.

---

## [Editor Report · Decision Letter 2]

Economic Assessment of Potential Changes to Essential Medicines for Diabetes in Uganda

PONE-D-25-01852R2

Dear Dr. Bonyani,

We’re pleased to inform you that your manuscript has been judged scientifically suitable for publication and will be formally accepted for publication once it meets all outstanding technical requirements.

Kind regards,

Sonak D. Pastakia

Academic Editor

PLOS ONE

Additional Editor Comments (optional):

Thank you for the prompt response and corrections.

Reviewers' comments:

Thank you for completing this important work.  I would encourage you to continue to refine and improve this framework and figure out ways to get this in the hands of other policy stakeholders in other countries who are grappling with the same decisions.  Please include glp-1's in future iterations.  I've thoroughly enjoyed reviewing this as it is very relevant to a lot of the work I'm currently doing in this domain and fills a major gap.

---

## [Editor Report · Acceptance letter]

PONE-D-25-01852R2

PLOS ONE

Dear Dr. Bonyani,

I'm pleased to inform you that your manuscript has been deemed suitable for publication in PLOS ONE. Congratulations! Your manuscript is now being handed over to our production team.

Kind regards,

on behalf of

Dr. Sonak D. Pastakia

Academic Editor

PLOS ONE